# Learning Unified Representations for Multi-Resolution Face Recognition

## Abstract

In this work, we propose Branch-to-Trunk network (BTNet), a novel representation learning method for multi-resolution face recognition. It consists of a trunk network (TNet), namely a unified encoder, and multiple branch networks (BNets), namely resolution adapters. As per the input, a resolution-specific BNet is used and the output are implanted as feature maps in the feature pyramid of TNet, at a layer with the same resolution. The discriminability of tiny faces is significantly improved, as the interpolation error introduced by rescaling, especially up-sampling, is mitigated on the inputs. With branch distillation and backward-compatible training, BTNet transfers discriminative high-resolution information to multiple branches while guaranteeing representation compatibility. Our experiments demonstrate strong performance on face recognition benchmarks, both for multi-resolution face verification and face identification, with much less computation amount and parameter storage. We establish new state-of-the-art on the challenging QMUL-SurvFace 1: N face identification task.

## 1 Introduction

Machine learning has advanced tremendously driven by deep learning methods, but is still severely challenged by various data specifications, such as data type, structure, scale and size, etc. For instance, face recognition (FR) is a well-established deep learning task, while the performance degrades dramatically in the testing domain that differs from the training one, influenced by factors of variance like resolution, illumination, occlusion, etc.

Most face recognition methods map each image to a point embedding in the common metric space by deep neural networks (DNNs). The dissimilarity of images can be then calculated using various distance metrics (e.g., cosine similarity, Euclidean distance, etc.) for face recognition tasks.

Recent advancements in margin-based loss (e.g., ArcFace Deng et al. (2019a), MV-Arc-Softmax Wang et al. (2020c), CurricularFace Huang et al. (2020), etc) enhanced discriminability of the metric space, with small intra-identity distance and large inter-identity distance. However, lack of variation in training data still leads to poor generalizability. Various useful methods are utilized to mitigate this issue. The model adapts to factors of variance by augmenting datasets, whereas the large discrepancy in data distribution could potentially weaken the model's ability to extract discriminative features with the same data scale and model structure (see Section 4.3). Fine-tuning is widely used to transfer large pretrained models to new domains with different data specifications. However, this strategy requires one to store and deploy a separate copy of the backbone parameters for every single new domain, which is expensive and often infeasible.

As known, the resolutions of face images in reality may be far beyond the scope covered by the model. As the small feature maps with a fixed spatial extent (e.g., $7 \times 7$) are mapped to an embedding with a predefined dimension (e.g., $128 - d$, $512 - d$, etc.) by a fully connected (fc) layer, input images need to be rescaled to a canonical spatial size (e.g., $112 \times 112$) before fed into the network. However, up-sampling low-resolution (LR) images introduces the interpolation error (see Section 3.1), deteriorating the recognizable ones which contain enough clues to identify the subject. Even though super-resolution methods (Zhu et al. (2016); Grm et al. (2020); Wang et al. (2016); Cheng et al. (2018a); Yin et al. (2020); Singh et al. (2019); Rai et al. (2020)) are widely used to build faces with good visualization, they inevitably introduce feature information of other identi-

Table 1: Correspondence between our goals and methods

|  | Compatibility | Discriminability |
|---|---|---|
| Input preprocessing | - | w/o rescaling to a canonical size |
| Network structure | TNet (unified encoder) | BNets (resolution adapters) |
| Training strategy | Back compatible training (influence loss) | Branch distillation |

ties when reconstructing high-resolution (HR) faces. This may lead to erroneous identity-specific features, which are detrimental to risk-controlled face recognition.

Empirically, we can divide inputs by resolution distribution and learn to operate on them via multiple models to achieve high accuracy and efficiency. However, multi-model fashion cannot be applied directly for cross-resolution recognition as representation compatibility among models need to be guaranteed (Shen et al. (2020); Budnik & Avrithis (2021); Wang et al. (2020a); Meng et al. (2021); Duggal et al. (2021)).

To improve discriminability while ensure the compatibility of the metric space for multi-resolution face representation, we learn the "unified" representation by a partially-coupled Branch-to-Trunk Network (BTNet). It is composed of multiple independent branch networks (BNets) and a shared trunk network (TNet). A resolution-specific BNet is used for a given image, and the output are implanted as feature maps in the feature pyramid of TNet, at a layer with the same resolution.

Furthermore, we find that multi-resolution training can be beneficial to building a strong and robust TNet, and backward-compatible training (BCT) Shen et al. (2020) can improve the representation compatibility during the training process of BTNet. To ameliorate the discriminability of tiny faces, we propose branch distillation in intermediate layers, utilizing information extracted from HR images to help the extraction of discriminative features for resolution-specific branches.

Our method is simple and efficient, which can serve as a general framework easily applied to existing networks to improve their robustness against image resolutions. Since multi-resolution face recognition is dominated by super-resolution and projection methods, to the best of our knowledge, our method is the first attempt to decouple the information flow conditioned on the input resolution, which breaks the convention of up-sampling the inputs. Meanwhile, BTNet is able to reduce the number of FLOPS by operating the inputs without up-sampling, and per-resolution storage cost by only storing the learned branches and resolution-aware BNs Zhu et al. (2021), while re-using the copy of the trunk model.

We demonstrate that our method performs comparably in various open-set face recognition tasks (1:1 face verification and 1: N face identification), while meaningfully reduces the redundant computation cost and parameter storage. In the challenging QMUL-SurvFace 1: N face identification task Cheng et al. (2018b), we establish new state-of-the-art by outperforming prior models.

In brief, our work can be summarized as follows: (1) **What is our goal?** Matching images with arbitrary resolutions (i.e., high-resolution, cross-resolution and low-resolution) effectively and efficiently, which is quite different from the traditional face recognition task. (2) **What is the core idea of our method?** Building unified (i.e.,compatible and discriminative) representations for multi-resolution images without introducing erroneous information. (3) **How to achieve our goal via our method?** Table 1 shows that we ensure the compatibility and discriminability from three aspects: input preprocessing, network structure, and training strategy.

## 2 RELATED WORK

**Compatible Representation Learning:** The task of compatible representation learning aims at encoding features that are interoperable with the features extracted from other models. Shen et. al. Shen et al. (2020) first formulated the problem of backward-compatible learning (BCT) and proposed to utilize the old classifier for compatible feature learning. Since the multi-model fashion benefits representation learning with lower computation, our idea of cross-resolution representation learning can be modeled similar to cross-model compatibility Shen et al. (2020); Budnik & Avrithis (2021); Wang et al. (2020a); Meng et al. (2021); Duggal et al. (2021), as metric space alignment

for different resolutions. Our goal is achieved by both compatibility-aware network architecture and training strategy.

**Knowledge Distillation and Transfer:** The concept of knowledge distillation (KD) was first proposed by Hinton et. al. in Hinton et al. (2015), which can be summarized as employing a large parameter model (teacher) to supervise the learning of a small parameter model (student). Distillation from intermediate features Heo et al. (2019a); Huang & Wang (2017); Park et al. (2019); Romero et al. (2015); Tung & Mori (2019); Zagoruyko & Komodakis (2016); Yim et al. (2017); Tian et al. (2019); Peng et al. (2019); Kim et al. (2018); Heo et al. (2019b) is widely adopted to enhance the effectiveness of knowledge transfer. However, due to the "dark knowledge" hidden in the intermediate layers, additional subtle design is often required to match and rescale intermediate features. Instead, our approach can easily locate the distillation features without rescaling and effectively transfer knowledge from the HR domain to LR branches.

**Low Resolution Face Recognition:** Its task includes low resolution-to-low resolution (LR-to-LR) matching and low resolution-to-high resolution (LR-to-HR) matching Martínez-Díaz et al. (2020). The work can be divided into two categories Luevano et al. (2021): (1) Super-resolution (SR) based methods aim to upscale LR images to construct HR images and use them for feature extraction Zhu et al. (2016); Grm et al. (2020); Wang et al. (2016); Cheng et al. (2018a); Yin et al. (2020); Singh et al. (2019); Rai et al. (2020). (2) Projection-based methods aim to extract adequate representations in different domains and project them into a common feature space Lu et al. (2018); Mudunuri et al. (2018); Zha & Chao (2019). SR approaches are able to build faces with good visualization, but inevitably introduce feature information of other identities when reconstructing corresponding HR faces, thus introducing noise for identity-specific features. Compared to previous projection methods, our approach directly learns discriminative representations in a common feature space for HR and LR inputs, without additional projection heads for feature transformation.

## 3 LEARNING SPECIFIC-SHARED FEATURE TRANSFER

Instead of rescaling the inputs to a canonical size, we build multiple resolution-specific branches (BNets) that are used to map inputs to intermediate features with the same resolution and a resolution-shared trunk (TNet) to map feature maps with different resolutions to a high-dimension embedding. We gain several important properties by doing so: (1) Processing inputs on its original resolution can diminish the inevitably introduced error via up-sampling or information loss via down-sampling, thus preserving the discriminability of visual information with different resolutions. (2) Information streams of different resolutions are encoded uniformly, thus enabling the representation compatibility, which is particularly beneficial to open-set face recognition considering that a compatible metric space is the prerequisite for computing similarity. (3) This also effectively reduce the computation for LR images by supplying computational resources conditioned on the input resolution.

### 3.1 UP-SAMPLING ERROR ANALYSIS

Figure 1 illustrates the experimental estimation of interpolation error, whose upper bound increases with the decline of the image resolution (see detailed theoretical derivation in Appendix A.1). Note that the error soars up when the resolution drops below 32 approximately which can be viewed as LR face images, consistent with the tiny-object criterion Torralba et al. (2008).

The results show that: (1) inputs with a resolution higher than around 32 can be considered in the same HR domain, since the error information introduced by up-sampling via interpolation can be ignored to a certain extent; (2) inputs with a resolution lower than around 32 should be treated as in various LR domains due to the high sensitivity of the resolution to errors.

### 3.2 BRANCH-TO-TRUNK NETWORK

Let $X$ be an input RGB image with a space shape: $X \in \mathbb{R}^{H \times W \times 3}$ where $H \times W$ corresponds to the spatial dimension of the input. For efficient batch training and inference, we predefine a canonical size $S \times S$ (e.g., $112 \times 112$ for typical face recognition models like ArcFace Deng et al. (2019a)).

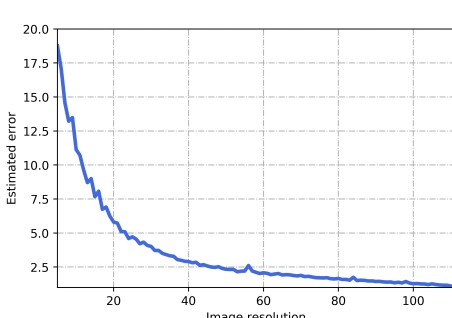

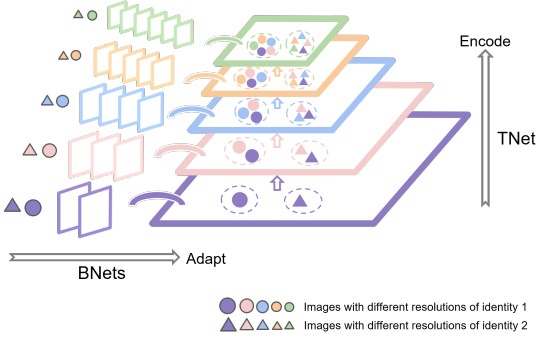

Figure 1: Estimated Error Upperbound. (bilinear interpolation, average value for over 100 images) with the change of image resolution relative to resolution 112.

Figure 2: Basic ideas of the proposed BTNet. Images of a certain identity are first projected to the feature maps with the same resolution respectively (Adapt) and then projected to a unified feature representation (Encode). In this figure, feature maps with the same resolution are indicated by outlines in the same color.

We build a trunk network $T : \mathbb{R}^{H \times W \times 3} \to \mathbb{R}^{C_{emb}}$ capable of extracting discriminative information with different resolutions, where $C_{emb}$ is the number of embedding channels. For every resolution $r$ in the candidate set, we formulate a resolution-specific branch, $z_r = B_r(X_r)$, which maps the input image $X_r$ to feature maps with the same resolution and expanded channels $z_r : \mathbb{R}^{r \times r \times 3} \to \mathbb{R}^{r \times r \times C_r}$. The idea is to learn our branches $B$ to focus on resolution-specific feature transfer independently. Feature maps will then be coupled to the trunk network $T$ in the feature pyramid with the same spatial resolution $r \times r$, allowing for further mapping to the unified presentation space by $T_r : \mathbb{R}^{r \times r \times C_r} \to \mathbb{R}^{C_{emb}}$.

Here, we follow the idea of "avoiding redundant up-sampling". Our branches $B$ are implemented with same-resolution mapping: i.e., the model preserves the network architecture of $T$ from input to the layer with resolution $r$ and abandons down-sampling operations (e.g., replacing the convolution of stride 2 with stride 1, abandoning the pooling layers, etc.) to keep the same-resolution flow.

We specifically name our specific-shared feature transfer network as Branch-to-Trunk Network, abbreviated as "BTNet". Figure 2 visually summarizes the main ideas of BTNet.

## 3.3 TRAINING OBJECTIVES

We now describe the training objectives. The training of BTNet includes training the trunk network $T$ such that it can produce discriminative and compatible representations for multi-resolution information, and fine-tuning the branch networks $B$ to encourage them to learn resolution-specific feature transfer, so as to improve accuracy without compromising compatibility.

**Influence Loss.** It is a compatibility-aware classification loss which is implemented by feeding the embeddings of the new model to the classifier of the old model Shen et al. (2020).Any classification-based loss (e.g., NormFace Wang et al. (2017), SphereFace Liu et al. (2017), CosFace Wang et al. (2018), ArcFace Deng et al. (2019a), etc.) can be refined as our influence loss, in the form of:

$$L_{influence} = L_{cls}(\varphi_{bt}, \kappa^*) \tag{1}$$

where $\varphi_{bt}$ is BTNet backbone (both $B_r$ and $T_r$), and $\kappa^*$ is the classifier of the pretrained trunk $T$.

**Branch Distillation Loss.** Due to the continuity of the scale change of both the image pyramid and the

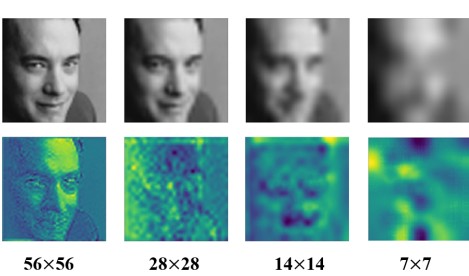

| 56×56 | 28×28 | 14×14 | 7×7 |

Figure 3: Visual comparison of face image-feature map pairs with different resolutions (resized to a common size here for illustration).

feature pyramid Lindeberg (1994), we can get a qualitative sense of the similarity between images and feature maps with the same resolution (see Figure 3). Furthermore, features extracted from HR images have richer and clearer information than those from LR images Lui et al. (2009). Motivated by these analyses, we utilize an MSE loss to encourage the branch output $z_r$ to be similar to the corresponding feature maps of the pretrained trunk network $z_s$:

$$L_{branch} = \frac{1}{V} \sum_{v=1}^{V} (z_{r_v} - z_{s_v})^2 \tag{2}$$

where $V$ denotes the batch size.

The whole training objective is a combination of the above objectives:

$$L = L_{influence} + \lambda_{branch} L_{branch} \tag{3}$$

where $\lambda_{branch}$ is a hyper-parameter to weigh the losses and we set $\lambda_{branch} = 0.5$ in all our experiments.



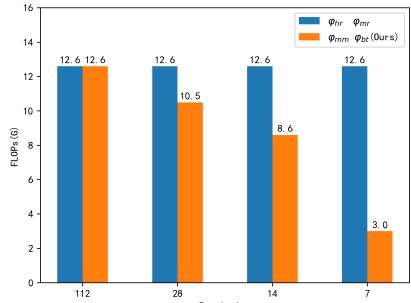

Figure 4: Comparison of # Params (M) between fully finetuning and $\varphi_{bt}$.

Figure 5: Comparison of FLOPs (G) between baselines and $\varphi_{bt}$.

### 3.4 STORING BRANCH NETWORKS

An obvious adaptation strategy is fully finetuning of the model on each resolution. However, this strategy requires one to store and deploy a separate copy of the backbone parameters for every resolution, which is an expensive proposition and difficult to expand into more segmented resolution branches. Our BTNet is beneficial in the scenario of multi-resolution face recognition which achieves better parameter/accuracy trade-offs. Since activation statistics including means and variances under different resolutions are incompatible Touvron et al. (2019), we update and store Batch Normalization (BN) Ioffe & Szegedy (2015) parameters in all layers of $B_r$ and $T_r$ for each resolution, whose amount is negligible. Apart from this, we only need to store the learned branches and re-use the original copy of the pretrained trunk model, significantly reducing the storage cost. Figure 4 shows that BTNet requires only $1.1\% \sim 48.9\%$ of all the parameters compared to fully updating all the parameters of TNet.

## 4 EXPERIMENTS

### 4.1 IMPLEMENTATION DETAILS

**Datasets.** We use MS1Mv3 Deng et al. (2019b) for training face embedding models. The MS1Mv3 dataset contains 5,179,510 images of 93,431 celebrities. We try on six widely adopted face verification benchmarks: LFW Huang et al. (2008), CFP-FF Sengupta et al. (2016), CFP-FP Sengupta et al. (2016), AgeDB-30 Moschoglou et al. (2017), CALFW Zheng et al. (2017), and CPLFW Zheng & Deng (2018), while the large-scale surveillance face dataset QMUL-SurvFace Cheng et al. (2018b) is used for 1:N face identification, which contains native LR surveillance faces across wide space and time. The spatial resolution for QMUL-SurvFace ranges from 6/5 to 124/106 in height/width with an average of 24/20.

**Baselines.** In our experiment, several baselines are used to validate BTNet in learning discriminative and compatible representations for multi-resolution face recognition.

·**High-Resolution Trained** $\varphi_{hr}$. Naive baseline trained with HR data.

·**Independently Trained** $\varphi_{mm}$. Multi-model fashion: is it possible to achieve better results if we train a specific model for each resolution independently? Specifically, we train $\varphi_r$ for data with resolution $r$ and denote the multi-model collections as $\varphi_{mm}$.

·**Multi-Resolution Trained** $\varphi_{mr}$. Trained with multi-resolution data which adapts to resolution-variance. For a comprehensive evaluation, we implemented three baselines, denoted as $\varphi_{mr}$, $\varphi_{mr(v2)}$, $\varphi_{mr(v3)}$ respectively. Each image is down-sampled to a certain size and then up-sampled to $112 \times 112$. The differences are as follows: (i)$\varphi_{mr}$: down-sampled to a size in the candidate set $\{\frac{112}{2^i} \times \frac{112}{2^i} | i = 0, 1, 2, 3, 4\}$ with equal probability of being chosen. (ii)$\varphi_{mr(v2)}$: down-sampled to a size in the candidate set with unequal probability of being chosen $[0.3, 0.25, 0.2, 0.15, 0.1]$. (iii)$\varphi_{mr(v3)}$: down-sampled to a size in the candidate interval $[4, 112]$.

**Instantiation of Network Architecture.** The BTNet and baselines are implemented with ResNet50 He et al. (2016), and they could be extended easily with other implementations. Dubbed as $\varphi_{bt}$, the detailed instantiation of BTNet based on ResNet50 is illustrated in Appendix A.2.

**Training.** The training details can be found in Appendix A.3.

## 4.2 EVALUATION METRICS

On the benchmarks for face verification, we use 1:1 verification accuracy as the basic metrics. The rank-20 true positive identification rates (TPIR20) at varying false positive identification rates (FPIR) and AUC are used to report the identification results on QMUL-SurvFace.

For better evaluation, we define another two metrics to assess the relative performance gain similar to Shen et al. (2020); Meng et al. (2021).

**Cross-Resolution Gain.** With the purpose towards the cross-resolution compatible representations, we define the performance gain as follows:

$$Gain_{r_1 \& r_2}(\varphi) = \frac{M_{r_1 \& r_2}(\varphi) - M_{r_1 \& r_2}(\varphi_{hr})}{|M_{r_1 \& r_2}(\varphi_{mr}) - M_{r_1 \& r_2}(\varphi_{hr})|} \tag{4}$$

Here $M_{r_1 \& r_2}(\cdot)$ are metrics when the resolutions of the image/template pair are $r_1 \times r_1$ and $r_2 \times r_2$ ($r_1 \neq r_2$), respectively. $\varphi_{mr}$ shares the same architecture with $\varphi_{hr}$ while is trained on multi-resolution images and thus serves as the baseline of cross-resolution gain.

**Same-Resolution Gain.** For the scenario of multi-resolution face recognition, the performance of same-resolution verification/identification is also vital besides cross-resolution one. Therefore, we report the relative performance improvement from base model $\varphi_{hr}$ in the scenario of same-resolution.

$$Gain_{r \& r}(\varphi) = \frac{M_{r \& r}(\varphi) - M_{r \& r}(\varphi_{hr})}{|M_{r \& r}(\varphi_r) - M_{r \& r}(\varphi_{hr})|} \tag{5}$$

Here $M_{r \& r}(\cdot)$ are metrics when the resolutions of the image/template pair are both $r \times r$. $\varphi_r$ is a model of the set $\{\varphi_{mm} = \varphi_r | r = 7, 14, 28\}$ trained on images with resolution $r \times r$ without considering cross-resolution representation compatibility, which serves as the baseline of same-resolution gain on resolution $r$. Note that for both metrics we add the absolute symbol to the denominator as they can be negative in some test settings (detailed in Section 4.3).

## 4.3 RESULTS

### 4.3.1 MULTI-RESOLUTION FACE VERIFICATION

We now conduct experiments on the proposed BTNet framework for multi-resolution identity matching. Two different settings are included : (1) same-resolution matching, and (2) cross-resolution matching. Table 2 compares the average performance on popular benchmarks for $\varphi_{hr}$, $\varphi_{mm}$, $\varphi_{mr}$, $\varphi_{bt}$.

When directly applied to test data with the resolution lower than training data, $\varphi_{hr}$ suffers a severe performance degradation. Up-sampling images via interpolation can increase the amount of data

Table 2: Comparison of different methods on six face verification benchmarks.

(a) Cross-resolution identity matching.

| | 112&7 | | 112&14 | | 112&28 | |
|---|---|---|---|---|---|---|
| | Acc. | Gain | Acc. | Gain | Acc. | Gain |
| $\varphi_{hr}$ | 57.75 | - | 81.02 | - | 95.90 | - |
| $\varphi_{mm}$ | 50.58 | -0.89 | 49.90 | -4.82 | 50.03 | -305.80 |
| $\varphi_{mr}$ | 65.85 | +1.00 | 87.47 | +1.00 | 96.05 | +1.00 |
| $\varphi_{mr(v2)}$ | 65.68 | +0.98 | 87.13 | +0.95 | 95.70 | -1.33 |
| $\varphi_{mr(v3)}$ | 68.80 | +1.36 | 88.13 | +1.10 | 96.62 | +4.80 |
| $\varphi_{bt}$(Ours) | **86.10** | **+3.50** | **94.08** | **+2.02** | **96.65** | **+5.00** |

(b) Same-resolution identity matching.

| | 7&7 | | 14&14 | | 28&28 | | 112&112 | |
|---|---|---|---|---|---|---|---|---|
| | Acc. | Gain | Acc. | Gain | Acc. | Gain | Acc. | Gain |
| $\varphi_{hr}$ | 60.70 | - | 73.88 | - | 93.58 | - | **97.68** | - |
| $\varphi_{mm}$ | 62.57 | +1.00 | 78.00 | +1.00 | 94.68 | +1.00 | **97.68** | - |
| $\varphi_{mr}$ | 61.02 | +0.17 | 80.32 | +1.56 | 95.12 | +1.40 | 97.25 | - |
| $\varphi_{mr(v2)}$ | 60.82 | +0.06 | 80.22 | +1.54 | 95.63 | +1.86 | 96.82 | - |
| $\varphi_{mr(v3)}$ | 61.62 | +0.49 | 80.55 | +1.62 | 94.78 | +1.09 | 97.52 | - |
| $\varphi_{bt}$(Ours) | **77.78** | **+9.13** | **90.90** | **+4.13** | **96.27** | **+2.45** | 97.25 | - |

but not the amount of information, only to improve the detailed part of the image and the spatial resolution (size) Liu & Liu (2003). Moreover, it also brings various noise and artificial processing traces Siu & Hung (2012). Up-sampling images via interpolation-typically bilinear interpolation or bicubic interpolation of 4x4 pixel neighborhoods, essentially a function approximation method, is bound to introduce error information (detailed in Appendix A.1), thus potentially confusing identity information, which is especially crucial for LR images with limited details. We are able to observe improvement of $\varphi_{mm}$ in same-resolution matching but its cross-resolution gain is negative with approximately 50% accuracy. Unsurprisingly, independently trained $\varphi_r$ is unaware of representation compatibility, and thus does not naturally suitable for cross-resolution recognition. The results show that $\varphi_{mr}$ improved both cross-resolution and same-resolution accuracy by a large margin, as it learns to adapt to resolution variance and maintain discriminability of multi-resolution inputs. Note that the model size and training data scale stay the same, while only the resolution distribution of the data changes for $\varphi_{mr}$, and thus there is a marginal accuracy drop in the setting of 112&112 matching. Comparably, $\varphi_{bt}$ substantially outperforms all baselines with 2.02 ~5.00 cross-resolution gain and 2.45~9.13 same-resolution gain. Importantly, due to the multi-resolution branches, our approach has a cost same with $\varphi_{mm}$, significantly lower than $\varphi_{hr}$ and $\varphi_{mr}$ (see Figure 5).

Moreover, we investigate the deviation in the accuracy change between different datasets, and assess the robustness of the face recognition systems to image resolutions.We can find that our proposed approach is much more robust than baselines against image resolution, and can also remain effective with more factors of variance (e.g., large pose variations, large age gap, etc.) included.

### 4.3.2 MULTI-RESOLUTION FACE IDENTIFICATION

In the native scenario, it is common to inference on inputs with resolutions not strictly matched to the branch. Since the low-quality image may possess an underlying optical resolution significantly lower than its size due to degraded quality caused by noise, blur, occlusion, etc Wong et al. (2010). , there exists dislocation between the underlying optical resolution of native face images and that of a branch. To avoid introducing extra large-scale parameters for predicting the image quality, three heuristic selection strategies based on different resolution indicators are validated (see Figure 8). Table 3 compares BTNet against the state-of-the-arts models on QMUL-SurvFace 1:N identification benchmark. We are able to observe that our proposed approach extends the state-of-the-arts while being more computationally efficient. We believe the performance of BT-Net (max + ceil) is the highest that have been reported so far, and we believe it is meaningful with the increased focus on unconstrained surveillance applications.

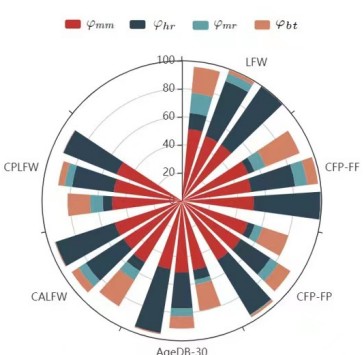

Figure 6: Detailed cross-resolution face verification comparison of different methods on six benchmarks for different image resolutions. The clockwise sequence indicates 112&7,112&14,112&28 matching per-benchmark.

## 5 ABLATION STUDY

Table 4: Comparison of different training methods for our BTNet. "Acc." denotes average 1:1 verification accuracy. "# Params." indicates the amount of parameter storage for the branch network $B_{14}$.

| Training method | Acc. (%) | | # Params. (M) |
|---|---|---|---|
| | 112&14 | 14&14 | |
| Scratch | 49.90 | 78.00 | 43.59 |
| Pretraining | 78.05 | 76.87 | 43.59 |
| Pretraining + BCT | 85.90 | 78.04 | 43.59 |
| Pretraining + BCT + Fix Trunk | 85.07 | 77.22 | 2.29 |
| Pretraining + BCT + Fix Trunk + Branch Distillation | 94.08 | 90.90 | 2.29 |

Table 5: Ablation study of different loss functions.

| Implementation of influence loss | 112&14 Acc.(%) | 14&14 Acc.(%) |
|---|---|---|
| CosFace | 94.10 | 90.78 |
| ArcFace | 94.17 | 90.88 |
| CurricularFace | 94.08 | 90.90 |

In all these experiments, we report the average verification results on six benchmarks in 112&14 and 14&14 matching, representing cross-resolution and same-resolution performance respectively.

**Training Method Alternatives.** Here, we experimentally compare different training methods: (1) Scratch: train without pretrained trunk parameters. (2) Pretraining: initialize the backbone and classifier with the pretrained trunk network. (3) Backward-compatible training (BCT, Shen et al. (2020)): fix parameters of the old classifier. (4) Fix-trunk: fix parameters of the trunk subnet $T_r$. (5) Branch distillation: use L2-distance to obtain the loss between the intermediate feature maps at the coupling layer of the pretrained trunk $T$ and the branch $B_r$.

We compare different training method combinations in Table 4 and find that both pretraining and BCT succeeded in ensuring representation compatibility. Among these two, BCT performs better since it imposes a stricter constraint during training. Furthermore, we are able to observe that branch distillation is crucial for improving the discriminative power by transferring high-resolution information to low-resolution branches.

**Loss Functions.** Since the difficulties of samples vary due to image resolution, we compute CurricularFace Huang et al. (2020) as our classification loss in the original architecture, which distinguishes both the difficultness of different samples in each stage and relative importance of easy and hard samples during different training stages.

To prove the main technical contribution of BTNet (rather than other components), we use different loss functions to replace the CurricularFace loss as influence loss in the original architecture. The comparison results(in Table 5) demonstrate that there is no

Table 3: Performance of face identification on QMUL-SurvFace. Most compared results are cited from Cheng et al. (2018b); Fang et al. (2020); Low & Teoh (2022), except AdaFace and BTNet.

| | TPIR20(%)@FPIR | | | |
|---|---|---|---|---|
| | AUC | 0.3 | 0.2 | 0.1 |
| VGG-Face Parkhi et al. (2015) | 14.0 | 5.1 | 2.6 | 0.8 |
| DeepID2 Sun et al. (2014) | 20.8 | 12.8 | 8.1 | 3.4 |
| FaceNet Schroff et al. (2015a) | 19.8 | 12.7 | 8.1 | 4.3 |
| SphereFace Liu et al. (2017) | 28.1 | 21.3 | 15.7 | 8.3 |
| SRCNN Dong et al. (2014) | 27.0 | 20.0 | 14.9 | 6.2 |
| FSRCNN Dong et al. (2016) | 27.3 | 20.0 | 14.4 | 6.1 |
| VDSR Kim et al. (2016) | 27.3 | 20.1 | 14.5 | 6.1 |
| DRRN Tai et al. (2017) | 27.5 | 20.3 | 14.9 | 6.3 |
| LapSRN Lai et al. (2017) | 27.4 | 20.2 | 14.7 | 6.3 |
| ArcFace Deng et al. (2019a) | 25.3 | 18.7 | 15.1 | 10.1 |
| RAN Fang et al. (2020) | 32.3 | 26.5 | 21.6 | 14.9 |
| SST Du et al. (2020) | - | 12.4 | - | 9.7 |
| MASST Shi et al. (2021) | - | 12.2 | - | 9.2 |
| MIND-Net Low et al. (2021) | 31.9 | 25.5 | - | 20.4 |
| AdaFace Kim et al. (2022) | 32.6 | 28.3 | 23.6 | 16.5 |
| BTNet (avg.+floor) | 32.6 | 27.9 | 23.4 | 16.5 |
| BTNet (avg.+near) | 34.6 | 30.3 | 25.7 | 18.9 |
| BTNet (avg.+ceil) | **35.4** | 31.1 | 26.8 | 20.3 |
| BTNet (min+floor) | 32.3 | 27.6 | 23.2 | 16.1 |
| BTNet (min+near) | 34.0 | 29.6 | 25.0 | 18.0 |
| BTNet (min+ceil) | 35.3 | 31.0 | 26.6 | 19.9 |
| BTNet (max+floor) | 33.6 | 29.1 | 24.5 | 17.6 |
| BTNet (max+near) | 35.2 | 31.0 | 26.4 | 19.6 |
| BTNet (max+ceil) | **35.4** | **31.2** | **26.9** | **20.6** |

significant difference among different implementations of influence loss. It means that the main performance gain is attributed to our novel design.

**Where should we have resolution-specific layers?** We conducted an ablation to see the effects of different specific-shared layer allocation strategies. The experiment was done with different trunk layers (i.e., the parameters of these layers are inherited from the pretrained trunk without updating). Figure 7 shows the results. We find that increasing the number of branch layers (i.e., specific layers for different resolutions) will lead to better performance due to increased flexibility. Our specific-shared layer allocation of BTNet can achieve better parameter/accuracy tradeoffs. Since further

increasing the number of trunk layers based on BTNet cannot lead to significantly better performance but increases parameter storage cost by a large margin, we use resolution-specific layers as shown in Figure 9.

## 6    DISCUSSION AND CONCLUSION

This paper works on the problem of multi-resolution face recognition, and provides a new scheme to operate images conditioned on its input resolution without large span rescaling. The error introduced by up-sampling via interpolation is investigated and analyzed. Decoupled as branches for discriminative representation learning and coupled as the trunk for compatible representation learning, our Branch-to-Trunk Network (BTNet) achieves significant improvements on multi-resolution face verification and identification tasks. Besides, the superiority of BTNet in reducing computational cost and parameter storage cost is also demonstrated. It is worth noting that our approach is easy to expand to recognition tasks for other classes of objects and has the potential to serve as a general network architecture for multi-resolution visual recognition.

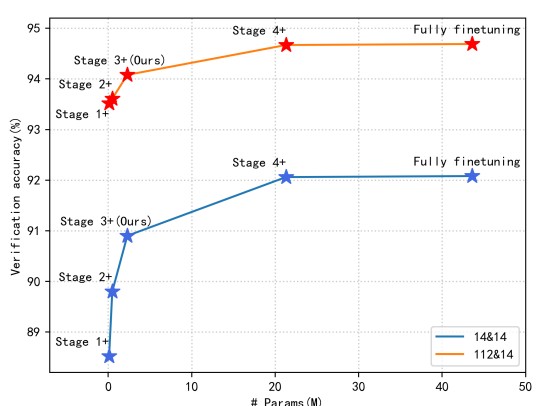

Figure 7: Comparison of verification accuracy and the amount of stored parameters for different specific-shared layer allocation strategies. Note that "Stage x+" indicates that layers deeper than "Stage x, Unit 1" are inherited from the pretrained trunk without updating.

**Limitations and Future Work.**    The dislocation between the underlying optical resolution of native face images and that of a certain branch may limit the power of the model, which may be improved by selecting the optimal processing branch for the input in combination with the image quality, rather than by image size alone. The optimal branch selection strategy is not fully investigated though we have provided an intuitive way to select the branch for inputs (see Figure 8). Importantly, based on the unified multi-resolution metric space, the underlying resolution of the inputs (integrated spatial resolution with quality assessment) can be utilized to provide the reliability of the representation and contribute to risk-controlled face recognition. They will be our future research directions.

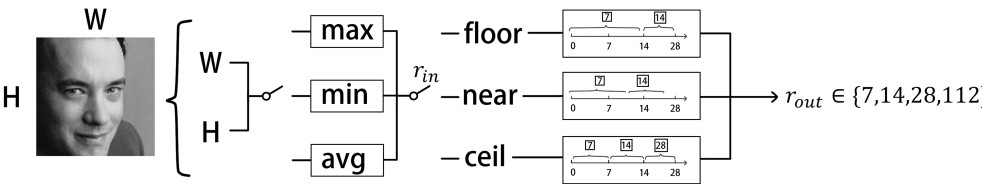

Figure 8: Branch selection process. Max/min/average is used on (W, H) to obtain a resolution indicator for further allocation (floor/near/ceil) to a certain branch.

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

# A APPENDIX

## A.1 THEORETICAL DERIVATION OF UP-SAMPLING ERROR

Here, we take bilinear interpolation, a typical image interpolation method, as an example to analyze the relationship between the interpolation error and the resolution of a face image. Bilinear interpolation can be considered as a bivariate Lagrange interpolation problem containing two interpolation nodes in each of the two dimensions.

Let $D$ be a unit-bounded closed region in a two-dimensional image space, and $Q_1(x_0, y_0)$, $Q_2(x_1, y_0)$, $Q_3(x_0, y_1)$, $Q_4(x_1, y_1) \in D$ be four adjacent pixel points in this region. We use an interpolation polynomial $P(x, y)$ for the interpolation approximation of the bivariate continuous function $f(x, y)$ defined on $D$, and the interpolation error $E(x, y)$ can be expressed as

$$E(x, y) = f(x, y) - P(x, y) \tag{6}$$

which indicates the potential error information introduced to the recognition of different identities. According to the the Rolle's theorem, we can obtain

$$E(x, y) = \frac{\frac{\partial^4 f(\xi, \eta)}{\partial x^2 \partial y^2}}{4} \omega_2(x) \mu_2(y) \tag{7}$$

where $\xi, \eta$ is an interior point of $D$ and

$$\omega_2(x) = (x - x_0)(x - x_1) \tag{8}$$

$$\mu_2(y) = (y - y_0)(y - y_1) \tag{9}$$

As $x_1 - x_0 = y_1 - y_0 = 1$ for adjacent pixel points, we can get the upper bound of $|\omega_2(x)|$ and $|\mu_2(y)|$

$$|\omega_2(x)| < \frac{1}{4}, |\mu_2(y)| < \frac{1}{4} \tag{10}$$

Thus, the error estimation can be expressed as

$$E(x,y) \le \frac{|\frac{\partial^4 f(\xi,\eta)}{\partial x^2 \partial y^2}|}{64} \tag{11}$$

where $\frac{\partial^4 f(\xi,\eta)}{\partial x^2 \partial y^2}$ can be approximated using the difference operator

$$\begin{bmatrix} 1 & -2 & 1 \\ -2 & 4 & -2 \\ 1 & -2 & 1 \end{bmatrix} \tag{12}$$

Based on the above theoretical analysis, we can experimentally study the relationship between the estimated up-sampling error and the image resolution.

### A.2 INSTANTIATION OF BTNET-RES50

We provide the detailed architecture of BTNet-res50 ($\varphi_{bt}$), an instantiation of BTNet framework based on ResNet50 He et al. (2016). Our method can be easily implemented by refining a network with the top-down hierarchical representation structure.

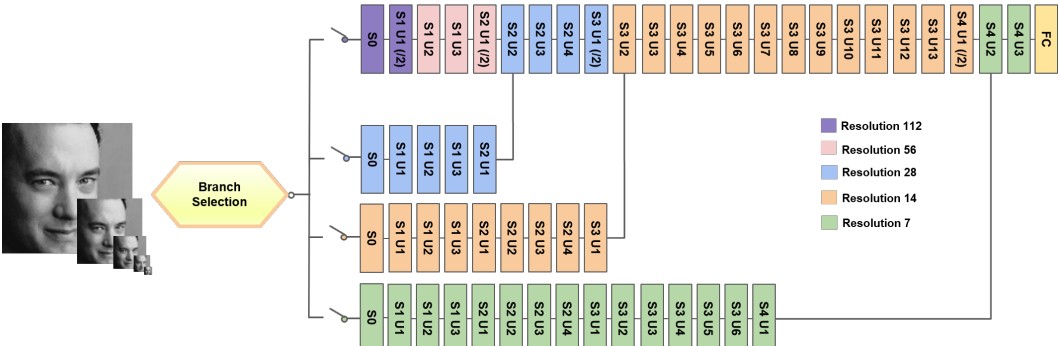

Figure 9: Detailed architecture of BTNet-res50 ($\varphi_{bt}$). Note that 'S' and 'U' represent stage and unit respectively, and '/2' means down-sampling by convolution with stride 2.

### A.3 TRAINING DETAILS

**Training.** All the models are trained on four RTX 2080 Tis with batch size 128 by stochastic gradient descent. For TNet, we train for 25 epochs, with learning rate initialized at 0.2 with 2 warm-up epochs and decaying as a quadratic polynomial. We augment training samples by random horizonal flipping and multi-resolution training. For BNets, we initialize the learning rate by 0.02 without warm-up epochs. The training all stops at the $10th$ epoch for a fair comparison. The recommended hyper-parameters are used for classification loss from the original paper (e.g., $m = 0.5, s = 64$ for ArcFace Deng et al. (2019a), and $\alpha = 0.99, t^0 = 0$ for CurricularFace Huang et al. (2020)). Only horizontal flipping is used as augmentation when training BNets.

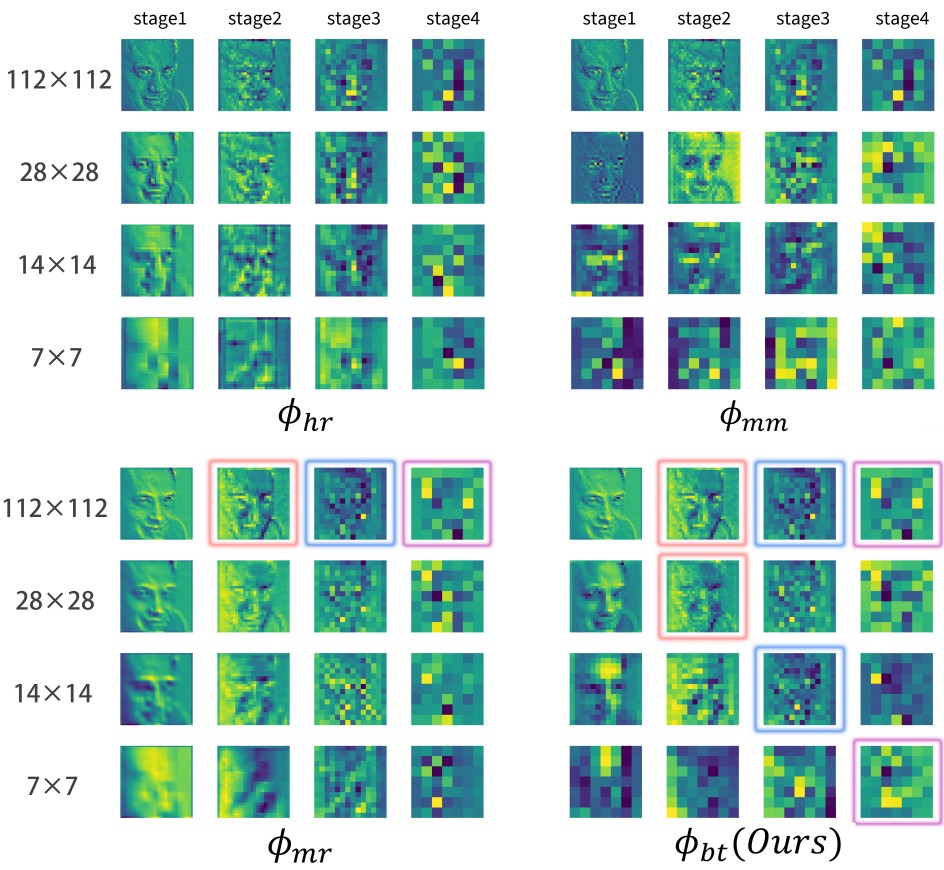

Figure 10: Visualization of intermediate feature maps for inputs with different resolutions. We show the feature maps located at output layers of BNets, denoted as stage1/2/3/4 respectively. We see our method can transfer multi-resolution visual inputs to intermediate feature maps at corresponding layers (indicated by bounding boxes of the same color) of TNet.

## A.4 VISUALIZATION

To interpret the behavior of learning compatible and discriminative representations, we visualize the intermediate feature maps in Figure 10. We find that $\varphi_{hr}$ introduces the noise information while $\varphi_{mm}$ has more discriminative but resolution-variant feature maps. The feature maps of $\varphi_{mr}$ tend to be smoother, diminishing the error information, but the discriminability could be limited as high-frequency details benefit recognition Wang et al. (2020b).

We also show that through the resolution-specific feature transfer of multiple branches, $\varphi_{bt}$ can encourage the transferred features to be aligned before fed into the trunk network in corresponding layers. For instance, at stage 2, the feature maps of $\varphi_{bt}$ with input resolution 112 and 28 are more similar than those of $\varphi_{hr}$, $\varphi_{mm}$, $\varphi_{mr}$. Furthermore, more detailed information can be found in the feature maps of $\varphi_{bt}$ with input resolution 28 compared to $\varphi_{mr}$. This inspiring phenomenon suggests that BTNet can learn compatible representations while improving the discriminability in low-resolution domain through the knowledge transferred from high-resolution visual signals.

## A.5 MORE EXPERIMENTS

Multi-resolution feature aggregation is common in set-based recognition tasks where the model needs to determine the similarity of sets (templates), instead of images. Each set could contain images of the same identity with different resolutions. In our experiment, we rescale the original and flipped images in each set to different resolutions and aggregate their features into a representation of the template.

Table 6: 1:1 verification TAR at different FAR on the IJB-C dataset for cross-resolution feature aggregation.

| | 112&7 | | | | | 112&14 | | | | | 112&28 | | | | |
|---|---|---|---|---|---|---|---|---|---|---|---|---|---|---|---|
| FAR | $10^{-6}$ | $10^{-5}$ | $10^{-3}$ | $10^{-2}$ | $10^{-1}$ | $10^{-6}$ | $10^{-5}$ | $10^{-3}$ | $10^{-2}$ | $10^{-1}$ | $10^{-6}$ | $10^{-5}$ | $10^{-3}$ | $10^{-2}$ | $10^{-1}$ |
| $\varphi_{hr}$ | **67.99** | **81.65** | 93.18 | 96.38 | 98.65 | 78.83 | 87.44 | 95.86 | 97.79 | 99.05 | **88.87** | **92.56** | 97.19 | 98.33 | 99.06 |
| $\varphi_{mm}$ | 53.57 | 64.34 | 84.01 | 91.96 | 97.12 | **83.22** | **89.56** | 96.10 | 97.71 | 98.82 | 86.84 | 92.33 | 97.16 | 98.10 | 99.01 |
| $\varphi_{mr}$ | 37.83 | 49.12 | 76.80 | 88.32 | 95.79 | 77.97 | 85.46 | 95.64 | 97.79 | 99.21 | 85.55 | 91.86 | 97.25 | 98.46 | 99.19 |
| $\varphi_{bt}$ (Ours) | 66.84 | 78.40 | **94.27** | **97.63** | **99.16** | 81.92 | 88.38 | **96.64** | **98.34** | **99.28** | 86.61 | 92.48 | **97.38** | **98.47** | **99.20** |

Table 7: 1:1 verification TAR at different FAR on the IJB-C dataset for same-resolution feature aggregation.

| | 7&7 | | | | | 14&14 | | | | | 28&28 | | | | | 112&112 | | | | |
|---|---|---|---|---|---|---|---|---|---|---|---|---|---|---|---|---|---|---|---|---|
| FAR | $10^{-6}$ | $10^{-5}$ | $10^{-3}$ | $10^{-2}$ | $10^{-1}$ | $10^{-6}$ | $10^{-5}$ | $10^{-3}$ | $10^{-2}$ | $10^{-1}$ | $10^{-6}$ | $10^{-5}$ | $10^{-3}$ | $10^{-2}$ | $10^{-1}$ | $10^{-6}$ | $10^{-5}$ | $10^{-3}$ | $10^{-2}$ | $10^{-1}$ |
| $\varphi_{hr}$ | 0.69 | 1.73 | 12.58 | 27.63 | 56.81 | 9.82 | 20.38 | 52.57 | 72.61 | 90.30 | 75.67 | 83.24 | 94.21 | 97.15 | 98.74 | **89.58** | **94.51** | **97.57** | 98.40 | 99.06 |
| $\varphi_{mm}$ | 0.68 | 1.73 | 11.93 | 27.48 | 56.84 | 7.59 | 15.61 | 48.28 | 71.13 | 91.04 | 73.68 | 85.14 | 95.82 | 97.65 | 98.89 | **89.58** | **94.51** | **97.57** | 98.40 | 99.06 |
| $\varphi_{mr}$ | 0.74 | 1.76 | 11.11 | 25.98 | 54.26 | 14.21 | 24.72 | 60.39 | 79.84 | 94.35 | 78.91 | 86.42 | 96.04 | 98.07 | 99.09 | 88.48 | 93.37 | 97.50 | **98.51** | **99.23** |
| $\varphi_{bt}$ (Ours) | **12.09** | **20.70** | **57.17** | **79.02** | **93.90** | **57.75** | **70.63** | **90.85** | **96.06** | **98.68** | **82.85** | **90.32** | **96.94** | **98.31** | **99.15** | 88.48 | 93.37 | 97.50 | **98.51** | **99.23** |

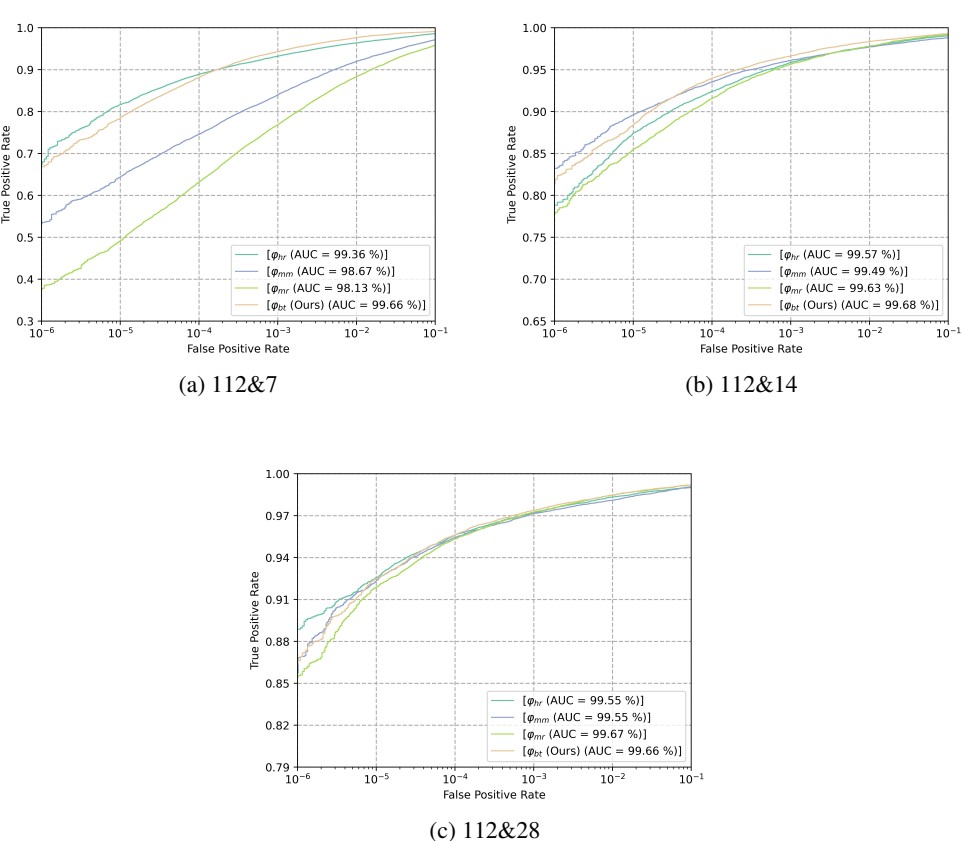

(a) 112&7

(b) 112&14

(c) 112&28

Figure 11: 1:1 verification ROC Curve on the IJB-C dataset for cross-resolution feature aggregation.

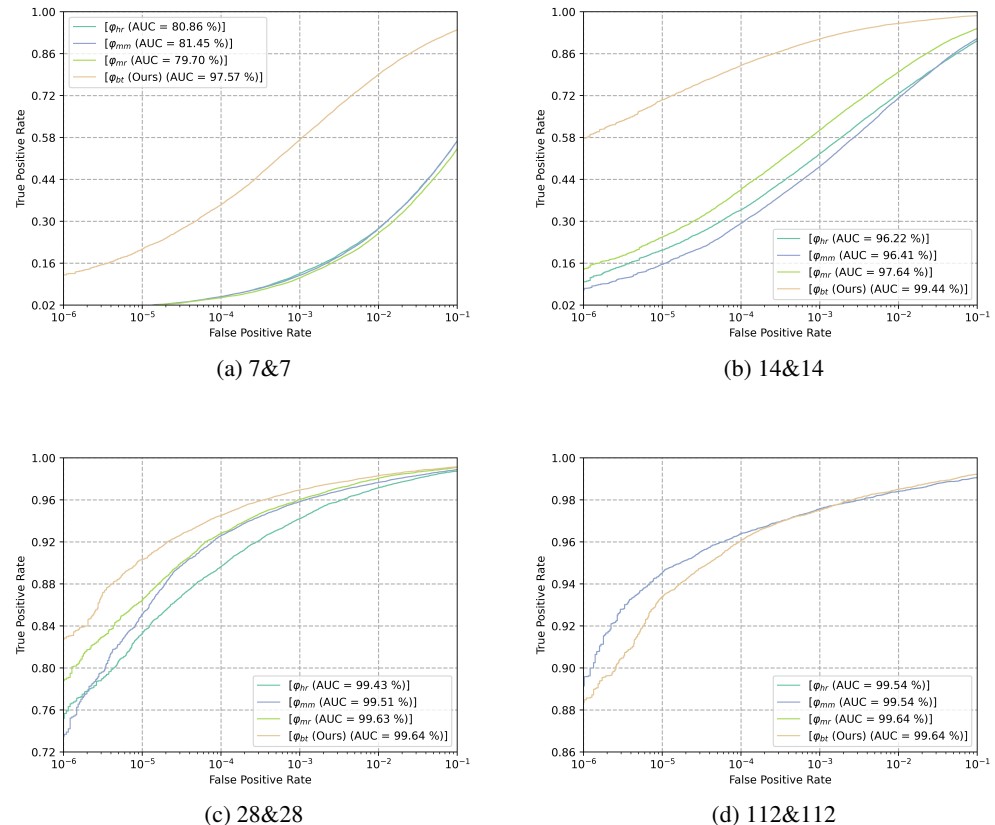

Figure 12: 1:1 verification ROC Curve on the IJB-C dataset for same-resolution feature aggregation.

Table 8: 1: N identification TPIR(%@FPIR=0.01), Top-1, Top-5, Top-10 accuracy on the IJB-C dataset for cross-resolution feature aggregation.

|  | 112&7 | | | | 112&14 | | | | 112&28 | | | |
|---|---|---|---|---|---|---|---|---|---|---|---|---|
|  | TPIR | Top-1 | Top-5 | Top-10 | TPIR | Top-1 | Top-5 | Top-10 | TPIR | Top-1 | Top-5 | Top-10 |
| $\varphi_{hr}$ | **75.35** | **92.76** | **95.14** | **95.92** | 81.98 | 93.89 | 96.25 | 96.98 | **90.42** | 96.05 | **97.47** | 97.80 |
| $\varphi_{mm}$ | 59.07 | 88.89 | 92.33 | 93.35 | **86.39** | **95.15** | **96.86** | 97.31 | 90.04 | 96.00 | 97.31 | 97.72 |
| $\varphi_{mr}$ | 43.89 | 82.29 | 87.74 | 89.42 | 82.18 | 93.87 | 96.20 | 96.89 | 88.90 | 95.93 | 97.36 | 97.84 |
| $\varphi_{bt}$(Ours) | 73.40 | 91.30 | 94.86 | 95.88 | 84.78 | 94.78 | 96.84 | **97.41** | 89.84 | **96.16** | 97.46 | **97.90** |

Table 9: 1: N identification TPIR(%@FPIR=0.01), Top-1, Top-5, Top-10 accuracy on the IJB-C dataset for same-resolution feature aggregation.

|  | 7&7 | | | | 14&14 | | | | 28&28 | | | | 112&112 | | | |
|---|---|---|---|---|---|---|---|---|---|---|---|---|---|---|---|---|
|  | TPIR | Top-1 | Top-5 | Top-10 | TPIR | Top-1 | Top-5 | Top-10 | TPIR | Top-1 | Top-5 | Top-10 | TPIR | Top-1 | Top-5 | Top-10 |
| $\varphi_{hr}$ | 1.20 | 11.77 | 19.95 | 24.28 | 15.16 | 50.96 | 63.62 | 68.68 | 77.52 | 91.62 | 94.95 | 95.99 | **92.66** | **96.58** | **97.71** | 97.94 |
| $\varphi_{mm}$ | 1.24 | 20.38 | 30.23 | 34.83 | 11.62 | 62.08 | 72.33 | 76.33 | 79.31 | 93.87 | 96.09 | 96.81 | **92.66** | **96.58** | **97.71** | 97.94 |
| $\varphi_{mr}$ | 1.36 | 17.41 | 26.53 | 31.03 | 23.72 | 68.64 | 78.38 | 81.99 | 83.82 | 94.53 | 96.67 | 97.33 | 90.89 | 96.44 | 97.65 | **98.00** |
| $\varphi_{bt}$(Ours) | **15.55** | **55.49** | **67.98** | **73.05** | **63.69** | **86.35** | **92.14** | **94.01** | **86.87** | **95.42** | **97.06** | **97.62** | 90.89 | 96.44 | 97.65 | **98.00** |

Table 10: Comparison of different methods on the IJB-C dataset 1:1 face verification task. "TAR" denotes TAR (%@FAR=1e-4).

(a) Cross-resolution feature aggregation.

| | 112&7 | | 112&14 | | 112&28 | |
|---|---|---|---|---|---|---|
| | TAR | Gain | TAR | Gain | TAR | Gain |
| $\varphi_{hr}$ | **88.89** | - | 92.40 | - | **95.62** | - |
| $\varphi_{mm}$ | 74.54 | -0.56 | 93.52 | +1.33 | 95.42 | -0.69 |
| $\varphi_{mr}$ | 63.11 | -1.00 | 91.56 | -1.00 | 95.33 | -1.00 |
| $\varphi_{bt}$(Ours) | 88.17 | -0.03 | **93.97** | **+1.87** | 95.62 | **+0.00** |

(b) Same-resolution feature aggregation.

| | 7&7 | | 14&14 | | 28&28 | | 112&112 | |
|---|---|---|---|---|---|---|---|---|
| | TAR | Gain | TAR | Gain | TAR | Gain | TAR | Gain |
| $\varphi_{hr}$ | 4.83 | - | 33.74 | - | 89.65 | - | **96.40** | - |
| $\varphi_{mm}$ | 4.83 | + 0.00 | 29.26 | -1.00 | 92.58 | +1.00 | **96.40** | - |
| $\varphi_{mr}$ | 4.48 | - | 40.51 | +1.51 | 92.81 | +1.08 | 96.06 | - |
| $\varphi_{bt}$(Ours) | **35.47** | - | **82.08** | **+10.79** | **94.50** | **+1.66** | 96.06 | - |

Table 11: Comparison of different methods on the IJB-C dataset 1: N face identification task. "TPIR" denotes TPIR (%@FPIR=0.1).

(a) Cross-resolution feature aggregation.

| | 112&7 | | 112&14 | | 112&28 | |
|---|---|---|---|---|---|---|
| | TPIR | Gain | TPIR | Gain | TPIR | Gain |
| $\varphi_{hr}$ | **85.60** | - | 90.11 | - | 94.27 | - |
| $\varphi_{mm}$ | 69.70 | -0.55 | 91.73 | +1.53 | 94.13 | -0.33 |
| $\varphi_{mr}$ | 56.64 | -1.00 | 89.05 | -1.00 | 93.84 | -1.00 |
| $\varphi_{bt}$(Ours) | 83.93 | -0.06 | **91.87** | **+1.66** | **94.33** | **+0.14** |

(b) Same-resolution feature aggregation.

| | 7&7 | | 14&14 | | 28&28 | | 112&112 | |
|---|---|---|---|---|---|---|---|---|
| | TPIR | Gain | TPIR | Gain | TPIR | Gain | TPIR | Gain |
| $\varphi_{hr}$ | 3.12 | - | 26.37 | - | 86.06 | - | **95.57** | - |
| $\varphi_{mm}$ | 3.24 | +1.00 | 21.84 | -1.00 | 89.76 | +1.00 | 95.57 | - |
| $\varphi_{mr}$ | 3.25 | +1.08 | 37.58 | +2.47 | 91.02 | +1.34 | 94.85 | - |
| $\varphi_{bt}$(Ours) | **27.70** | **+204.83** | **76.65** | **+11.10** | **92.89** | **+1.85** | 94.85 | - |

Table 10 (a) compares the cross-resolution results of TAR@FAR=$10^{-4}$ for 1:1 verification. The cross-resolution features are ensured to be mapped to the same vector space where the aggregation is conducted for $\varphi_{hr}$ and $\varphi_{mr}$, but we can observe that $\varphi_{hr}$ performs much better than $\varphi_{mr}$. One possible reason is that $\varphi_{hr}$ has outstanding discriminability to extract HR features, while LR features may not overly deteriorate the HR information. This phenomenon also suggests that $\varphi_{mr}$ sacrifices its discriminability in exchange for the adaptability for resolution-variance. We can see $\varphi_{bt}$ is comparable with $\varphi_{hr}$, demonstrating the discriminative power of BTNet for aggregating multi-resolution features.

Table 10 (b) compares the same-resolution results of TAR@FAR=$10^{-4}$ for 1:1 verification. When HR information is removed from the template representation (i.e., test settings 7&7, 14&14, 28&28), $\varphi_{hr}$ suffers from performance degradation as well, as the informative embedding cannot catch the lost details of the LR images Fang et al. (2020). Both $\varphi_{mm}$ and $\varphi_{mr}$ improve with a limited same-resolution gain, while $\varphi_{bt}$ surpasses the baselines by a large margin while also reducing the compute.

In Table 11 we show the results of TPIR@FPIR=$10^{-1}$ for 1:N identification protocol. Similar to our results for 1:1 verification, we are able to observe that $\varphi_{bt}$ is comparable or even better than $\varphi_{hr}$ with HR information involved and can preserve superior discriminability with limited LR information, while also being more computationally efficient.

We report the detailed results on the IJB-C dataset, including TAR at different FAR (see Table **??**, 6), ROC Curve (see Figure 11, 12) for 1:1 verification, and TPIR at FPIR=0.01, Top-1, Top-5, Top-10 accuracy (see Table 8, 9) for 1:N identification. We are able to observe that $\varphi_{bt}$ can be comparable to or serve as the paradigm model (i.e., model with the best performance) in each resolution setting, both for identity matching and feature aggregation.

