# OpenReview forum: "Learning Unified Representations for Multi-Resolution Face Recognition"
_ICLR.cc/2023/Conference — Submitted to ICLR 2023_

### Official Review · Reviewer_FLmy · 2022-10-24

**Confidence:** 4
**Correctness:** 4
**Technical Novelty And Significance:** 3
**Empirical Novelty And Significance:** 2
**Recommendation:** 5

**Clarity, Quality, Novelty And Reproducibility:**

Clarity: The paper is well presented, logical and clear, with standard citations.

Quality: This article is of average quality and is not recommended for acceptance.

Novelty: I feel that this article is not innovative enough.

Reproducibility: This article may be reproduced.


**Strength And Weaknesses:**

Strength: The design of the model structure, but the pyramid structure is already very common in other work and cannot be considered as a very innovative point.

Weakness:
There are several points in the article that confused me, as follows:
1. In Table 4, the number of parameters of Pretraining + BCT is 43.59, while the number of parameters of Pretraining + BCT + Fix Trunk is 2.29. It seems unreasonable.
2. As can be seen from Table 4, it is the model distillation operation that provides the largest performance improvement across multi-resolution and same resolution task. What puzzles me is: 1) whether distillation is capable of such a significant improvement and 2) if the distillation operation can improve to such an extent, then the innovation points claimed in this paper may not be reliable and more gains come from the distillation operation.
3. The task of this paper is to obtain a better cross-resolution representation, and I think it may be more intuitive to visualize it using t-SNE.



**Summary Of The Paper:**

In this paper, the author designed a new model,named Branch-to-Trunk network (BTNet), to match faces with different resolutions, and the experimental result demonstrates that the method is feasible. A possible potential contribution of this paper is the design of the network structure, mainly the introduction of multiple branches.

**Summary Of The Review:**

  The quality of this article is average and not recommended for acceptance.

---

### Official Review · Reviewer_JeWQ · 2022-10-24

**Confidence:** 4
**Correctness:** 2
**Technical Novelty And Significance:** 2
**Empirical Novelty And Significance:** 2
**Recommendation:** 3

**Clarity, Quality, Novelty And Reproducibility:**

The paper is not easy to read. It could not be reproduced with the information provided. However, some coded has been provided as supplementary material.

**Strength And Weaknesses:**

Strengths.

The paper addresses a relevant problem in the literature, namely, multi-resolution face recognition.

Weaknesses.

The experimentation needs to be improved. It only compares with the state-of-the-art in a low-resolution data set. If it it claims to learn a multi-resolution representation, it should be compared with other competing algorithms and on many more data sets with different resolutions/characteristics.

The writing should also be improved. The paper is difficult to understand. Important information is missing, e.g. how is the representation produced by BTNet combined with the backbone. In section 4.1 it reads that the  experimentation has been performed with 6 data sets that go missing in the paper. Some tables at the end of the paper were not cited in the text.


**Summary Of The Paper:**

The paper presents a multi-resolution face recognition algorithm. It is based on a CNN encoder backbone (e.g. ResNet50), denoted trunk Net, and multiple lateral resolution-specific nets, denoted BTNets. Each BTNet receives as input an image with a specific resolution producing a representation of the same resolution. This representation is combined with the backbone at the layer with a resolution matching that of the BTNet. The experimentation shows how different components of the model contribute to the final solution. It compares favorably with the state of the art on a low resolution data set.

**Summary Of The Review:**

The paper addresses a relevant problem in face recognition. However, the presentation of the approach is difficult to follow and is not complete. The experimentation is weak. It compares with the state of the art in only one data set, failing to evaluate its multi-resolution performance against its competitors.

---

### Official Review · Reviewer_Lmwt · 2022-10-25

**Confidence:** 4
**Correctness:** 3
**Technical Novelty And Significance:** 2
**Empirical Novelty And Significance:** 3
**Recommendation:** 5

**Clarity, Quality, Novelty And Reproducibility:**

The clarity and novelty of this paper is not good enough. The reproducibility of this paper is good.

**Strength And Weaknesses:**

Strength:
Most low-resolution face recognition works require up-sampling low-resolution images to a fixed size (112*112). This paper proposes a multi-branch network to reduce the interpolation error by piecewise processing the input image according to the resolution, which is a good idea. Meanwhile, this method only stores the learned branches and resolution-aware BNs, which requires less computation amount and parameter storage.

Weaknesses:
1. For the training set MS1Mv3, we need to first down-sample the high-resolution image to the low-resolution. Although this paper avoids up-sampling by branching strategy, down-sampling will still introduce interpolation errors and there is no solution for this question.

2. For the test on real low-resolution face images, the model does not know the specific resolution to select the branch. If the face resolution needs to be judged according to the image size, many images of QMUL-SurvFace are not full face (after face alignment), and the image collected in real scene may have background and need face detection. But low-resolution face detection is difficult to achieve. At this time, there is no judgment strategy for face resolution, which make it difficult to select branches.

3. This paper inputs images with different resolutions by adding branch headers in front of the backbone network, which is not novel enough.

4. The paper lacks experiments compared to the SOTA methods, such as 1:N face identification on SCFace dataset, 1:1 face verification on QMUL-SurvFace dataset and 1:N face identification on QMUL-TinyFace dataset.

5. In Table 2, the paper does not provide results for each of the six datasets. Tables 4 and 5 do not indicate which dataset the experiment was performed on. In figure 8, further allocation (floor/near/ceil) lacks a specific explanation.


**Summary Of The Paper:**

The paper proposes a Branch-to-Trunk network for multi-resolution face recognition. In this paper, by setting multiple branches and inputting images with different resolutions into different branches, the interpolation error caused by up-sampling is reduced. It requires less computation amount and parameter storage. The paper conducts experiments on six face verification benchmarks and QMUL-SurvFace dataset.

**Summary Of The Review:**

See Strength And Weaknesses.

---

### Official Review · Reviewer_yRHk · 2022-10-25

**Confidence:** 4
**Clarity, Quality, Novelty And Reproducibility:** The quality and clarity is good, and …
**Correctness:** 3
**Technical Novelty And Significance:** 2
**Empirical Novelty And Significance:** 3
**Recommendation:** 5

**Strength And Weaknesses:**

Strength:
+ The paper is well written and easy to read.
+ The proposed BTNet obtains the state-of-the-art performance on the challenging QMUL-SurvFace 1: N face identification task.

Weaknesses:
- Waht are contributions of the proposed BTNet compared to existing dynamic resolution network (Zhu et al., NeurIPS 2021)? And the paper should compare with dynamic resolution network.
- The paper evaluate the proposed BTNet on the challenging QMUL-SurvFace dataset, and it is recommend to conduct comparisons on more fae recognition benchmarks, e.g., WebFace260M [1].
[1] Zhu et al., WebFace260M: A Benchmark Unveiling the Power of Million-Scale Deep Face Recognition, CVPR 2021.

**Summary Of The Paper:**

This paper introduces a Branch-to-Trunk network (BTNet) for multi-resolution face recognition, which consists of a trunk network (TNet) and multiple branch networks (BNets). With branch distillation and backward compatible training, BTNet transfers discriminative high-resolution information to multiple branches. Experiments on face recognition benchmarks show the better performance of the proposed BTNet for multi-resolution face verification and face identification.

**Summary Of The Review:**

The main concerns of the work are its weaknesses as
- Waht are contributions of the proposed BTNet compared to existing dynamic resolution network (Zhu et al., NeurIPS 2021)? And the paper should compare with dynamic resolution network.
- The paper evaluate the proposed BTNet on the challenging QMUL-SurvFace dataset, and it is recommend to conduct comparisons on more fae recognition benchmarks, e.g., WebFace260M [1].

---

### Decision · Program_Chairs · 2023-01-20

**Decision:**

Reject

**Justification For Why Not Higher Score:**

Reviewers raised some valid concerns on contribution, experiments, and methodology, and they have consistent negative ratings. The authors did not provide any responses to the comments/questions from reviewers.

**Justification For Why Not Lower Score:**

N/A

**Metareview: Summary, Strengths And Weaknesses:**

This paper presents a Branch-to-Trunk network (BTNet) for multi-resolution face recognition. The proposed method consists of a trunk net and multiple lateral resolution-specific nets, denoted BTNets.

Overall, the paper is well organized and easy to follow. The proposed method is well motivated, and some experimental results on benchmarks are promising.

However, compared with existing work, the technical contributions of this paper is not significant. The experiments are not very convincing, since SOTA methods are only compared on a low-resolution dataset. Moreover, some details in methodology and experiments require further clarifications, as pointed out by reviewers. Hopefully these issues could be addressed in the next version of this work.

**Summary Of Ac-Reviewer Meeting:**

N/A